# Access to Health-Related Information, Health Services, and Welfare Services among South and Southeast Asian Immigrants in Japan: A Qualitative Study

**DOI:** 10.3390/ijerph191912234

**Published:** 2022-09-27

**Authors:** Sadatoshi Matsuoka, Madhu Kharel, Kyoko Koto-Shimada, Maiko Hashimoto, Hiroyuki Kiyohara, Azusa Iwamoto, Mika Nishihara, Masami Fujita

**Affiliations:** 1Bureau of International Health Cooperation, National Center for Global Health and Medicine, 1-21-1 Toyama, Shinjuku-ku, Tokyo 162-8655, Japan; 2Department of Community and Global Health, Graduate School of Medicine, The University of Tokyo, Bunkyo-ku, Tokyo 113-8655, Japan; 3Global Health Policy Division, International Cooperation Bureau, Ministry of Foreign Affairs, 2-2-1 Kasumigaseki, Chiyoda-ku, Tokyo 100-8919, Japan; 4Graduate School of Biomedical Sciences, Nagasaki University, 1-7-1 Sakamoto, Nagasaki 852-8521, Japan

**Keywords:** COVID-19, migrant health, access, health-related information, health services, welfare services

## Abstract

Migrants face several challenges in their daily lives in the host country due to limited knowledge about the language, culture, and social system of the host country. Their vulnerability increases in a time of crisis. During the COVID-19 pandemic, migrant communities were severely affected. Evidence on migrants’ access to COVID-19-related information and services is limited. We conducted a qualitative, descriptive study among migrants from Vietnam, Myanmar, and Nepal living in Japan to explore the barriers and promoting factors for their access to health-related information, health services, and welfare services during the first wave of COVID-19. We used a thematic analysis to identify key themes according to the study’s objectives. Further, these themes were assessed using an adapted version of the ecological model. The migrants mainly relied on the information available on social networking sites and were not aware of formal sources of information. Language was a major barrier, followed by cognitive bottlenecks and time constraints for migrants accessing health-related information and services. Social media, short-form information provided using their native language or plain Japanese and illustrations, and supportive people around could help them to access health-related information and services. The findings from this study demonstrate how migrants can represent a vulnerable group in a host country, even more so in a time of crisis.

## 1. Introduction

A certain proportion of international migrants represent a marginalized group in host countries, even during normal times [1]. They often face many challenges in their daily lives due to their limited knowledge of the language, culture, and system of the host country [2,3], and they are even more vulnerable during a time of crisis [4]. Providing migrants with accurate information and support services is crucial during such a time.

The COVID-19 pandemic that started in Wuhan, China in late 2019 has spread worldwide and affected many people through several waves [5,6]. As of 24 February 2022, it had resulted in 429.2 million infections and 5.92 million deaths globally [7]. When COVID-19 was first identified, many aspects of the infection were unclear, and the leading knowledge was rapidly evolving [8]. During the initial days, people were greatly fearful, and COVID-19 became the talk of the town. An overwhelming amount of information about COVID-19 was available on the internet and social media, leading to a COVID-19 infodemic [9,10,11]. It was difficult for people to access and select accurate information, including information on relevant health and welfare services.

In Japan, the rapid spread of COVID-19 infection led the government to declare a state of emergency on 7 April 2020, limiting people’s movement and leading to the closure of non-essential businesses [12]. This action resulted in a reduction in income or a total loss of work for many people [13,14]. The government tried to support people through various welfare services such as cash transfers (Special Cash Payment, JPY 100,000), housing allowances (Housing Support Benefit), and loans (Temporary Loan Emergency Fund, JPY 200,000; General Support Fund, up to JPY 600,000) [15]. They disseminated information on these services widely through news and media outlets and local advertising. Considering international migrants as a vulnerable group at the time of the pandemic, the government set up foreign residents’ information centers to provide relevant information regarding COVID-19 and help migrants access health and welfare services provided by the government [16]. However, it is unclear what proportion of the international migrants knew of and had access to the information and services provided by the government. In that context, we set out to explore the promoting factors and barriers of accessing health-related information and health and welfare services for international migrants in Japan. Further, these factors were assessed using an adapted version of the ecological model [17] to help identify promising points of intervention.

## 2. Materials and Methods

### 2.1. Design

We took a qualitative, descriptive approach, using a traditional content analysis, to fulfill the objective of this study.

### 2.2. Participants

Thirty-four participants in their 20s and 30s (and their dependents) were selected by the purposive sampling of foreign students studying at Japanese-language schools or vocational/technical schools, technical intern trainees, and cooks for Indian and Nepalese restaurants originating from Vietnam, Myanmar, or Nepal but living in Japan (Table A1). Purposive sampling was applied, since the population that is of specific of interest can be studied [18]. The participants were selected, since these three nationalities were among the fastest-growing foreign communities in Japan, and young adults in their 20s and 30s account for a large proportion of the migrants from these communities [19]. Purposive sampling was undertaken by postgraduate students of these nationalities in Tokyo, who reached out to their community networks and also contributed to data collection as assistants. The selection and recruitment of participants was undertaken through convenience, snow, and self-selection samplings. The students contacted their acquaintances who met the inclusion criteria, directly requested that they participate in the study, and/or requested the acquaintances/participants to refer further potential participants. In addition, information on the study was posted mainly on Facebook groups for recruitment.

### 2.3. Data Collection

Semi-structured individual interviews (IIs) and focus group discussions (FGDs) were conducted online from September to November 2021.

An interview guide for the IIs and FGDs was developed by the authors by adapting the interview guides developed by Matsuoka et al., aiming to identify barriers to the use of health services [20] and intending to elucidate challenges and constraints in the development and implementation of the regulatory framework for nursing professionals with a viewpoint of the ecological model in the Mekong region [21]. Then, the guide was translated into Vietnamese, Burmese, and Nepali by the assistants, back-translated by different assistants, pilot-tested, and finalized before data collection. In addition, prior to the development of the interview guide, the authors had consultation meetings with researchers and managers of support organizations familiar with these nationalities residing in Japan. IIs and FGDs were conducted primarily in English by two authors who had experience conducting research and/or providing technical assistance for projects in the health sector in low- and middle-income countries. In each II and FGD, two assistants supported the authors as interpreters. The IIs and FGDs were recorded on a digital voice recorder, transcribed by the assistants, checked, and finalized by the first author.

### 2.4. Ethical Considerations

This study was approved by the Ethical Committee of the National Center for Global Health and Medicine, Japan (no. 3632). All participants were informed that their data would be kept confidential, and they agreed to the audio recording of their responses.

### 2.5. Data Analysis

The authors undertook a thematic analysis by coding and categorizing key phrases from the transcripts, according to the study objectives, using a *MAXQDA 2018 version 18.2.5* (Berlin, Germany VERBI Software, 2021, maxqda.com, accessed on 15 March 2019). Barriers to accessing health-related information and health and welfare services were elucidated. In addition, opportunities and driving forces for addressing these barriers were identified. Furthermore, the barriers and driving forces identified were categorized based on an adapted version of the ecological model. This model proposes six levels of factors affecting health behavior: intrapersonal/individual factors (e.g., individuals’ knowledge, skills, behavior, attitude, values), interpersonal factors (e.g., individuals’ social networks, families, peers, neighbors), organizational factors (e.g., workplace, educational institutions), community factors (e.g., autonomous body, social and health services, cultural groups), public policy factors (e.g., legislation, policies), and societal factors (e.g., culture, customs, technological innovation) [17]. The initial coding, the categorization of main themes and subthemes, and the interpretations were primarily undertaken by the three authors (S.M., K.K.-S., and M.K.) who had experience in leading qualitative research projects in low- and middle-income settings. The remaining authors were briefed on the results of the primary analyses, and all the authors then discussed these together to explore each main theme and subtheme that could be inferred and interpreted the findings in the context of the literature, frequently involving a further review of the transcripts.

### 2.6. Rigor and Trustworthiness

Triangulation aims to enhance the credibility of qualitative research by applying multiple approaches to widen the scope of the work. Methodological triangulation was used here for data collection by applying IIs and FGDs [22]. Researcher triangulation was also undertaken at the stages of data collection, analysis, and interpretation. During the IIs and FGDs, the interviewers (S.M. and M.H.) noted keywords and phrases, framed them in appropriate sentences, and confirmed with the participants that they were correct and recorded with mutual consent [23]. In addition, the three authors who conducted the preliminary analyses (S.M., K.K.-S., and M.K.) briefed the remaining authors on the initial findings and interpretations, which they checked against the citations and the raw data to enhance the credibility of the research [24]. Beyond this, data triangulation was undertaken by collecting data from a variety of participants to enhance the credibility of the trends identified [25]. The triangulation approaches applied in this work contributed to enhancing the validity of the conclusions drawn, as well as our thorough description of all research processes. The transferability of the findings was also supported by our iterative sampling, our concurrent data analyses, and the mutual consensus on the theoretical saturation by all the authors [26]. The audio recordings, field notes, and transcription supports have further been stored so that all of the study’s findings can be verified, if needed [27].

## 3. Results

The IIs and FGDs enabled us to identify barriers to and facilitators of access to health-related information and health and welfare services.

### 3.1. Access to Health-Related Information

Barriers of language, cognition, and time were found. According to the ecological model, these barriers were categorized into individual and community levels. Similarly, facilitators were categorized into interpersonal, organizational, community, and societal levels (Table A2).

#### 3.1.1. Individual-Level Barriers

The most frequently cited language barrier to accessing health-related information was limited Japanese language proficiency. Migrants noted that information was provided mostly in Japanese, which was difficult for them to understand. A technical intern commented:


*“Official websites are complicated and are mainly in Japanese and English.”*
(Myanmar, technical intern)

Attempts to limit stress were identified as a cognitive barrier. Some of the respondents avoided accessing COVID-19-related information, since negative information on COVID-19 made them unhappy.


*“Initially, I was following the COVID-19-related news in a proactive way, but soon the COVID-19-related problems heated up more. Then, I realized that the more I look at the news, the more I became stressed. So, I decided to stop viewing those pages.”*
(Nepal, Japanese language student)

Most of the participants pointed out that time constraints were a barrier to accessing health-related information because they were busy working and studying.


*“We have to go to school and then work so we are a bit busy. We do not have time to know about the daily number of COVID-19 infections. So, we do not know how COVID-19 is affecting the country currently.”*
(Nepal, vocational/technical student)

#### 3.1.2. Community-Level Barriers

As a result of having limited Japanese language proficiency, the participants rarely used official sources of information, such as the official websites of the council or government, because the information was often provided in difficult Japanese. A technical student commented:


*“As for formal sources like official websites of the municipal or ward office. I rarely check them since the information provided is usually in difficult Japanese so it is not easy to understand.”*
(Myanmar, vocational/technical student)

Another barrier to accessing health-related information was the volume of information. Migrants knew that an enormous amount of information was available, and this prevented them from understanding the information properly, even when the information was provided in their own language.


*“Those Facebook*
*(FB) pages are quite popular and shared among my FB friends. It is mostly easy to understand and useful. But sometimes it is wordy and doesn’t directly mention [the information I am looking for,] and [given how much information is shared, I find it] quite difficult to catch up.”*
(Myanmar, technical intern)

A cognitive barrier at this level was the limited awareness of foreign residents’ information centers. Plenty of information was available on what to do when you suspected you had COVID-19, but migrants were unaware of this because the information had not reached them properly. Foreign residents’ information centers were established to provide COVID-19-related information and support foreign residents. However, few participants were aware of these information centers. Similarly, the migrants were unfamiliar with public health centers, although these centers played an important role in monitoring and referring infected COVID-19 patients.


*“The students do not know much about the COVID-19 consultation centers [Foreign residents’ information centers]. They know that some consultation centers exist but do not know how to access them.”*
(Nepal, vocational/technical student)

#### 3.1.3. Interpersonal- and Organizational-Level Facilitators

Those around the participants, including Japanese people, were also important sources of information.


*“I mostly learn information through my manager because he often comes and talks to me about COVID-19.”*
(Vietnam, technical intern)


*“[I got much of my information about COVID-19] from people in my community, including Japanese customers.”*
(Nepal, cook)

#### 3.1.4. Community-Level Facilitators

The main sources of health-related information for all three nationalities were SNSs, particularly Facebook (but also TikTok and YouTube). Migrant communities of each nationality have access to Facebook groups with news and important information for daily life in Japan. This is provided in their native language, so it is easier for the migrants to understand the information.


*“I have joined the Burmese people’s FB group in Japan because I want to learn about Japan, student life here, and Burmese organizations here. These groups provide news in Burmese so it is easy to understand.”*
(Myanmar, vocational/technical student)

Participants reported that information provided in plain Japanese (yasashii nihongo) with illustrations and animations was helpful for them. Plain Japanese uses all three components of the Japanese writing system: hiragana, katakana, and kanji Chinese characters, but at a level of second- or third-grade elementary school students. For the ease of understanding, hiragana can also appear above Chinese characters, called furigana, to indicate their pronunciation. Difficult terms are often rephrased. For example, an evacuation shelter would be “a place where everyone can stay for safety” (plain Japanese key to inclusive, multicultural Japan, available online: https://english.kyodonews.net/news/2020/01/04a67072447f-feature-plain-japanese-key-to-inclusive-multicultural-japan.html, accessed on 29 July 2022). A student studying the Japanese language mentioned:


*“The pamphlets from Kuyakusho [the local administrative office] are in Japanese, but the illustrations make them easy to understand”*
(Myanmar, Japanese-language student)

#### 3.1.5. Societal-Level Facilitators

Participants benefitted from technological evolution. Some migrants found language translation tools helpful in understanding health-related information provided in Japanese.


*“If I have to understand formal notices posted in the Japanese language, I use Google Translate, or I ask my teachers.”*
(Nepal, vocational/technical student)

### 3.2. Access to Health Services

Language, cognitive, and time barriers were found. Using the ecological model, these barriers were categorized into individual, organizational, and community levels. Facilitators were categorized into interpersonal and organizational levels (Table A2, red-highlighted text).

#### 3.2.1. Individual-Level Barriers

In addition to the issue of Japanese language proficiency, two types of cognitive barriers were observed: fear of infection with COVID-19 and perceived financial burden.

Fear of infection with COVID-19, and a negative impact on work or studies as a result, also prevented migrants from seeking healthcare.


*“No, I didn’t go [to any hospital] for the fever… If we go there, the hospital staff will test us to see if it is COVID-19, and the process will be lengthy. It will also affect our work and studies. Plus, if I didn’t have COVID-19 in the first place, I could contract it [from someone else at the hospital].”*
(Nepal, Japanese language student)

Perceived financial burden: some participants did not access health services because they suspected that they could be expensive; for that reason, they preferred to use commercially available medicines to cure their illnesses.


*“We don’t usually go to a health facility since we are afraid of spending money… When I feel like I am getting sick, I take the medicine that I brought in Myanmar in advance.”*
(Myanmar, Japanese language student)

Some participants mentioned time constraints on their ability to seek treatment or apply for health insurance.


*“Due to time constraints and all, I have not got Hoken [insurance] yet. I will get it soon. Initially, I had it, but I never had time to renew it after moving to another place [administrative area].”*
(Nepal, cook)

#### 3.2.2. Organizational-Level Barriers

Since technical health terms are difficult to understand for lay people in any language, the particular need to translate them into the participants’ mother languages or English was often mentioned.


*“Because the medical terms used in the hospital are very specialized, I think it is necessary to have a Vietnamese staff member to help and translate.”*
(Vietnam, technical intern)

#### 3.2.3. Community-Level Barriers

Migrants were unfamiliar with the Japanese healthcare system. They had no idea what to do if they suspected they had COVID-19.


*“I still don’t know what type of hospitals we can get coronavirus treatment from and where to go for the treatment.”*
(Nepal, cook)

#### 3.2.4. Interpersonal- and Organizational-Level Facilitators

Those who had accessed health services recalled having communicated with healthcare professionals using plain Japanese.


*“When they [the hospital staff] saw my residence card, they knew that I was a foreigner. So, they used Yasashii Nihongo [plain Japanese] to communicate and speak in an easy-to-understand manner”*
(Vietnam, vocational/technical student)

Japanese people who were close to participants in their daily lives, such as their bosses, company colleagues, and schoolteachers, were mostly mentioned as the sources of their support in accessing health services.


*“In the beginning, when I had a fall and picked up an injury, the tencho [branch manager] took me to the hospital. He accompanied me 2–3 times. By then, I had gotten used to it. From then on, I didn’t need the tencho to accompany me.”*
(Nepal, Japanese-language student)


*“If the students suspect they have COVID-19, first, they will inform their class teacher.”*
(Nepal, vocational/technical student)

Technical interns also mentioned support from their supervising organization.


*“I would inform the Kumiai [cooperative association] so that they can make arrangements for me.”*
(Myanmar, technical intern)

### 3.3. Access to Welfare Services

Barriers of language and cognition were found. Based on the ecological model, these barriers were categorized into individual and community levels. Facilitators were categorized into interpersonal, organizational, and community levels (Table A2, green-highlighted text).

#### 3.3.1. Individual-Level Barriers

As observed above, limited Japanese language proficiency was a barrier to easily accessing welfare services, leading to a limited awareness of welfare services available for participants at the individual level. The individual barriers are, in part, attributable to the community-level factors below.

#### 3.3.2. Community-Level Barriers

Many participants mentioned the Special Cash Payment (JPY 100,000) from the government for COVID-19. The need for help with translating the application forms came up.


*“Because my friend was not fluent in Japanese, when she filled out the form from the Shi [City Hall] to register, she mistakenly selected the box that said she did not need 10 Man (100,000 Yen). After that, I had to call the Shi and check on her application. Then, they sent another copy for her to do it again.”*
(Vietnam, technical intern)

Apart from the Special Cash Payment, participants’ knowledge about other types of governmental financial support was limited because the information had not reached them properly.


*“I did not know about any benefits other than the 100,000 Yen. I want to know more and where to get such information.”*
(Myanmar, technical intern)

#### 3.3.3. Interpersonal- and Organizational-Level Facilitators

Support from their workplace or supervising organization in applying for the governmental benefit was cited.


*“In April, we applied for the Special Cash Payment through the post, and our agent helped us. We didn’t have any difficulties.”*
(Myanmar, technical intern)


*“My manager showed us a sample application form, and he told us to fill it out like that, and then he also sent them off for us.”*
(Vietnam, vocational/technical student)

#### 3.3.4. Community-Level Facilitators

In addition, some mentioned that the local government officials also supported them with application procedures by explaining it them in plain Japanese.


*“Even with the application for 200,000 Yen [from the Temporary Loan Emergency Fund], the application procedure went well. The officer from the ward office explained it to me in easy-to-understand Japanese and helped me with filling out the forms, so I did not have any problems.”*
(Myanmar, vocational/technical student)

## 4. Discussion

This study identified barriers and facilitators for accessing health-related information and health and welfare services among South and Southeast Asian migrants in Japan during the first wave of the COVID-19 pandemic and categorized them using the ecological model. Language was a major barrier to accessing health-related information and health and welfare services, and it was identified at multiple levels of the ecological model, namely, the individual, organizational, and community levels. The language and cognitive barriers observed at the individual level, such as a limited awareness of social, health, and welfare services, were attributable to community-level factors.

Language was the major barrier to accessing health-related information, attributable to community- and individual-level factors. Most of the information provided by the government (community level) was in Japanese. Migrants had difficulty understanding such information. Only the students at Japanese-language schools and vocational/technical schools could understand plain Japanese. In a study from Denmark, Brønholt and colleagues also reported that language was a barrier to accessing health information about COVID-19 among migrants [28]. The large volume of information was another language barrier. The other barrier to accessing health-related information was a low awareness of the formal sources of information on social, health, and welfare services arising from community-level factors, which might have been a result of the language barrier [29]. Beyond this, at the individual level, some migrants refrained from accessing COVID-19-related information to avoid stress. Finally, time constraints presented another barrier to accessing health-related information. The participants did not have enough time to search for the required information.

However, at the interpersonal and organizational levels, supportive people around them such as their bosses and customers facilitated immigrants’ access to health-related information. At the community level, social media usefully facilitated their access to health-related information, with Facebook being the main source of information for all migrant communities. Beyond this, the migrants we sampled found language translation tools such as Google Translate and information delivered using plain Japanese and illustrations helpful for understanding the health information. Language translation tools are becoming increasingly handy to overcome language barriers, although their accuracy may be limited [30]. Social networking sites (SNSs) such as Facebook, TikTok, and YouTube are popular among migrants. They are easily accessible, easy to use, and serve multiple purposes such as connecting with families and friends and receiving news and entertainment [31]. We must note, however, that while SNSs may be easy and quick sources of information, the quality of information available on SNSs can be questionable, because its accuracy is rarely verified [32]. It is the responsibility of the readers to judge what they think is correct and take only that information on board. During the pandemic, people were confused by an overwhelming amount of conflicting information [33]. Such confusion hindered the government’s efforts to control the pandemic. We propose that short-form information culturally tailored to a migrant audience using their native language or plain Japanese and illustrations, posted on an SNS by concerned and reliable authorities and support organizations, e.g., on a popular Facebook page, would be an effective way to inform a large number of migrants [34,35].

During the COVID-19 pandemic, in Japan, migrants’ access to health services was largely hindered by multilevel factors. The language barrier was observed at the organizational level as well as the individual level. The migrants we sampled had trouble explaining their symptoms to the service providers because they felt that the medical terms in Japanese were complex. They wished someone were there who could have translated them into their native language or used plain Japanese. Several studies have reported that language is a key barrier to migrants accessing health services [36,37,38,39,40]. The migrants we sampled also had poor knowledge of the Japanese health system, which was caused by community factors, as the information from concerned authorities had not reached them properly. They were unsure of what they were meant to do if they had symptoms of COVID-19 and fell back on treating their symptoms using commercially available medicines rather than visiting a health facility, fearing that the Japanese healthcare services would be expensive. In Japan, most medical expenditures are, in principle, covered by the universal health insurance system (70–90%, depending on the age group). There is no charge for a COVID-19 test if one is required, at the discretion of a medical doctor (an initial consultation fee is charged), and if the patient tests positive for COVID-19, their medical costs are also free. However, migrants have arrived in this new place with an unfamiliar setup, and they perceive the system as complicated [41]. As an individual-level factor, the migrants we sampled feared getting COVID-19 if they visited a health facility. It would have a resulting impact on their work and/or studies, as they would have to isolate for several days. In addition, the migrants we sampled also commented that they did not have adequate time to seek health services. Several studies have reported time constraints as a barrier to accessing healthcare services among migrant populations [42,43]. Despite these challenges, migrants accepted help from people around them, such as their bosses, co-workers, schoolteachers, and the staff of supervising organizations, in seeking health services when they were in need.

Language and a low awareness of the government’s support measures were the barriers to accessing welfare services among the migrants. Most of the information about welfare services was provided in Japanese, and migrants with poor Japanese language ability had difficulty understanding the information. Migrants were also unaware of the various support measures provided by the government, except for the cash handout of JPY 100,000. In most cases, the migrants were unfamiliar with the support system of their host country [44]. However, as with health services, their access to welfare services was again facilitated to some extent by the support they received from their workplace and supervising organization, along with local government officials.

This study had some limitations that we must now note. First, we used purposive sampling to select the participants. While we tried to include variations within each group, since we were relying on the community networks of postgraduate students of the chosen nationalities in Tokyo, we could not reach any technical interns from Nepal (likely also related to their low number). In future studies, sampling through support organizations could be a more effective approach to take. A second point to note is that the IIs and FGDs were conducted online, as face-to-face meetings were restricted by the government’s state of emergency declaration to control the spread of the virus. While the effect was not the same as that of a face-to-face meeting, in most cases, we used video calls (when the participants agreed) for the interviews and discussions so that we could take participants’ expressions into account. Despite the limitations presented here, this is one of only a few studies to have explored migrants’ access to health-related information and health and welfare services during the early period of the pandemic, when little was known about the virus and there was great fear and uncertainty among the public.

## 5. Conclusions

The migrants we sampled mainly relied on information available on social networking sites and were not aware of many formal sources of information. They had some knowledge of the common preventive measures for COVID-19 infection. However, they were less prepared regarding what to do if they had symptoms of COVID-19. Language was the major barrier to accessing health-related information and health and welfare services among migrants. Their access to information and services was also limited by a lack of awareness about the support systems and procedures in their host country. Short-form information provided in their native language or plain Japanese and with illustrations, helped the migrants to overcome the language barrier to some extent. They also received support from people and organizations around them to facilitate their access to health and welfare services.

The findings from this study demonstrate how migrants represent a vulnerable group in the host country, even more so in a time of crisis. Providing relevant information concisely, in plain language, and using illustrations could be helpful for them. Moreover, supportive people and organizations around migrants could be mobilized to help them better access health and welfare services.

## Data Availability

The data presented in this study are available on request from the corresponding author. The data are not publicly available, as that would compromise personal information.

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
