# Peer review of "Access to Health-Related Information, Health Services, and Welfare Services among South and Southeast Asian Immigrants in Japan: A Qualitative Study"

_ijerph, 2022, doi:10.3390/ijerph191912234_

Round 1

Reviewer 1 Report

Conceptually weak paper that assumes that migrant populations are marginalised which is not always the case (and this assumption can increase stigmatisation) . It is not clear how this paper adds to the existing literature. Insufficient details about the selection of participants provided (what was the purpose of the purposive sampling?). No justification provided for the data collection tools used. Data obtained is thin and does not add much to existing knowledge. A conceptual framework should be used for a more insightful data analysis.

Author Response

Please see the attachment for comment 1-6.

Reviewer 2 Report

There are two concerns that I had reading this survey. Ignorance is not the only challenge during pandemic, but also false information. How much of that was caused by lack of information in the specific languages?

The second concern is abuse. Do we know if immigrants were blamed or harassed for spreading the pandemic?

I am aware that those concerns are beyond the scope of the research, but I wonder why.  

Author Response

Please see the attachment for comment 6 and 7.

Round 2

Reviewer 1 Report

Significant improvement but findings remain rather thin. A justification must be provided for the choice of methods. More specific details needed about the choice and process of selecting and recruiting participants and rational for the choice of data collection tools (the years of expertise of the researchers involved is not a justification of then choice of data collection tools)

Author Response

For the comment 1) above, we added the following explanations in the section 2.2 Participants (page 2):

"Selection and recruitment of participants was undertaken through convenience, snow, and self-selection samplings. The students contacted their acquaintances who met the inclusion criteria, directly requested them to participate in the study, and/or requested the acquaintances/participants to refer further potential participants. In addition, information on the study was posted mainly on Facebook groups for recruitment."

For the comment 2) above, we added the following explanations in section 2.3 Data Collection (page 3):

"(An interview guide for the IIs and FGDs was developed by the authors) by adapting the interview guides developed by Matsuoka et al. aiming at identifying barriers to use of health services (Matsuoka, 2010) and intended to elucidating challenges and constraints in the development and implementation of the regulatory framework for nursing professionals with a viewpoint of the ecological model in the Mekong region (Matsuoka, 2021). "